# Triggering Degradation of Host Cellular Proteins for Robust Propagation of Influenza Viruses

**DOI:** 10.3390/ijms25094677

**Published:** 2024-04-25

**Authors:** Chuan Xia, Ting Wang, Bumsuk Hahm

**Affiliations:** 1Department of Microbiology, College of Basic Medical Sciences, Dalian Medical University, Dalian 116044, China; 2Department of Bioengineering, College of Life Science and Technology, Jinan University, Guangzhou 510632, China; wangting1988@jnu.edu.cn; 3Departments of Surgery & Molecular Microbiology and Immunology, University of Missouri, Columbia, MO 65212, USA

**Keywords:** influenza virus, virus-host interaction, protein degradation, interferon, ubiquitination

## Abstract

Following infection, influenza viruses strive to establish a new host cellular environment optimized for efficient viral replication and propagation. Influenza viruses use or hijack numerous host factors and machinery not only to fulfill their own replication process but also to constantly evade the host’s antiviral and immune response. For this purpose, influenza viruses appear to have formulated diverse strategies to manipulate the host proteins or signaling pathways. One of the most effective tactics is to specifically induce the degradation of the cellular proteins that are detrimental to the virus life cycle. Here, we summarize the cellular factors that are deemed to have been purposefully degraded by influenza virus infection. The focus is laid on the mechanisms for the protein ubiquitination and degradation in association with facilitated viral amplification. The fate of influenza viral infection of hosts is heavily reliant on the outcomes of the interplay between the virus and the host antiviral immunity. Understanding the processes of how influenza viruses instigate the protein destruction pathways could provide a foundation for the development of advanced therapeutics to target host proteins and conquer influenza.

## 1. Introduction

Influenza viruses are the pathogens that cause influenza and belong to the Orthomyxoviridae family. There are four currently known types of influenza viruses: influenza A, B, C, and D. Among them, the influenza A virus (IAV) infects the most diverse host species, including humans, birds, swine, horses, and other mammals, and is most likely to cause a pandemic [1,2]. Influenza A viral genome consists of eight segments of negative-sense viral RNAs that encode at least 11 proteins, including viral nucleoprotein (NP) and polymerase proteins (PA, PB1, and PB2), which form the viral ribonucleoprotein (vRNP) complex together with the genomic RNA. vRNP complexes are surrounded by viral matrix protein 1 (M1) and a lipid bilayer derived from the host cell as an envelope. Two types of glycoproteins, hemagglutinin (HA) and neuraminidase (NA), are embedded in the envelope. There is also viral non-structural protein 1 (NS1), which is well-established as a key interferon antagonist [3] and also displays other functions beneficial for viral propagation [4,5,6], NS2 protein, which is also known as NEP (nuclear export protein), viral matrix protein 2 (M2) which is present on the virion as an ion channel, and a multifunctional protein PB1-F2 which participates in multiple processes of viral pathogenicity [7,8,9,10].

As acellular organisms, influenza viruses need to parasitize within the host cells in order to self-replicate, propagate, and spread. To gain a better environment to effectively multiply, influenza viruses often hijack cellular mechanisms. As a result, viruses use or manipulate the host’s nucleic acid, protein, and other biochemical processes, enabling them to provide support for viral growth. However, the host has also developed various antiviral strategies to restrict the viruses. Therefore, the interaction between influenza viruses and their host is extremely complicated. Viruses have evolved to equip themselves with many strategies to hijack the host proteins (or signaling pathways) and combat the antiviral system. One of the most effective ways is to deliberately deplete the host’s crucial antiviral factors through protein degradation pathways.

It has been well-established that infection of cells with influenza virus leads to a broad decline in the synthesis of cellular proteins, which is known as the host Shut-Off [11,12]. This phenomenon influences numerous events/signaling inside the entire cell. However, it seems to be not sufficient to help viruses thoroughly overcome the host’s innate and adaptive antiviral immunity. Therefore, it is not surprising that the influenza virus actively employs different modes of action to down-regulate or even directly induce the degradation of specific host proteins that display antiviral activities for their own benefit.

In this review, we provide an overview of current knowledge about influenza virus-induced degradation of cellular proteins, highlighting the modes by which the viruses co-opt the host by degrading these proteins to support viral propagation.

## 2. Influenza Virus Life Cycle

To better understand the interplay between the influenza virus and its host, we first scan the life cycle of this virus. The cycle can be divided into several stages as follows: adsorption to the target cell; virus entry into the cell; vRNPs transport into the nucleus; synthesis of viral genome and protein; virion assembly and budding at the host cell plasma membrane (Figure 1) [7,8].

When influenza virion attaches to the target cell, viral HA binds to the sialic acid (SA) linked to galactose on the surface of the cell’s membrane [13]. There are two linkages between SA and galactose, which are α(2,3) and α(2,6). These linkages are critical for the specific binding of HA to different host species [14]. It is generally considered that human influenza viruses only recognize α(2,6), whereas avian and equine influenza viruses recognize α(2,3) linkage [15]. The viruses from swine recognize both, which makes swine a perfect mixing vessel for human and avian influenza viruses, giving the risk of severe pandemics [16]. However, studies have shown that some avian IAVs also bind to α(2,6) SA linkage. Wan et al. have determined that three H12 IAVs isolated from wild waterfowl hosts could bind to both α(2,6) and α(2,3) SA-linked receptors [17]. Guan et al. have shown that certain H7 IAVs also bind to both types of receptors with a higher preference for α(2,3) linkage [18]. The hosts of IAV also express species-specific SA derivatives, e.g., N-glycolylneuraminic acid (NeuGc), which is expressed in pigs and horses but not in humans. Studies revealed that avian, human, and equine viral HAs bind NeuGc as well as N-acetylneuraminic acid (NeuAc) [19]. After binding to the host cell’s sialic acid residues, the virus penetrates and enters the cell via receptor-mediated endocytosis. A low pH value triggers the fusion of the endosomal membrane with the viral envelope, releasing viral vRNP into the host cell’s cytoplasm with the help of viral ion channel protein M2 [7,20]. Unlike many other RNA viruses, the influenza virus replicates in the nucleus. Viral NP, PA, PB1, and PB2, which are the components of vRNP, all have nuclear localization signals (NLSs). They are capable of binding to the cellular nuclear import machinery, helping the vRNP enter the nucleus [21]. The replication of the influenza virus in the nucleus is a complex process. The virus is known to hijack the host’s transcriptional machinery for its own benefit in many ways, and it has been reviewed by many researchers [22,23,24]. In the viral genome, five open reading frames (ORFs) encode for five proteins individually. Segments 7 and 8 encode for two proteins each due to differential splicing, wherein segment 7 encodes for M1 and M2; segment 8 encodes for NS1 and NS2. There is another nonstructural protein encoded by the PB1 gene segment from an alternative ORF called PB1-F2, which has multiple functions through interacting with different cellular components, contributing to viral pathogenicity. Influenza viruses utilize the host’s splicing machinery to express their proteins while, in turn, preventing the host cell from using its own machinery for processing cellular mRNAs. The newly synthesized vRNP are exported from the nucleus to the cytoplasm with the help of NP, M1, and NS2 proteins. NP interacts with the cellular protein CRM1, which is critical for the export of vRNP. M1 directly binds to vRNP through the C-terminal end and, at the same time, binds to NS2 through the N-terminal portion, while NS2 binds to CRM1, facilitating the vRNP export out of the nucleus [21,25]. After vRNP export, the newly synthesized viral components will be packaged into progeny virions and be released out of the cell. As an enveloped virus, the influenza virus utilizes the host cell’s plasma membrane to form its own envelope, which contains viral HA, NA, and M2 on the surface. The last step of the influenza virus life cycle is to leave the host cell’s plasma membrane. At this step, NA cleaves the sialic acid residues from glycoproteins and glycolipids, helping the viral particles release from the host. The whole process of the influenza virus life cycle is very complicated, and there are still areas we do not understand thoroughly.

## 3. Influenza Viral Degradation of Cellular Proteins Involved in Interferon (IFN) Responses

### 3.1. Viral Degradation of Cellular Factors That Regulate Type I IFN Production Pathway

To establish an effective infection, the influenza virus needs to overcome the host’s diverse antiviral immune responses. Type I interferon (IFN-I) system is the first line of host defense against influenza virus infection [26,27]. It is rapidly induced upon infection and restricts viral replication and the spread of viruses [28]. Upon influenza virus infection, the cellular pathogen sensor, RIG-I, is primarily activated by recognition of the viral RNA (vRNA) in epithelial cells [29,30]. The downstream adaptor, mitochondria-anchored protein, MAVS is then engaged, followed by recruitment of IκB kinase ε (IKKε) and TANK-binding kinase 1 (TBK1), both of which are able to activate interferon regulatory factor (IRF) 3 and IRF7 by phosphorylation. Phosphorylated IRF3/7 then enters the nucleus, driving the production of IFN-β and variant IFN-α. IRF3 is responsible for the initial transcription of IFN-I, while IRF7 is important for the secondary wave of IFN-I production [31,32]. After secreted, IFN-I binds to the cognate receptor (IFNAR) and activates the so-called JAK-STAT signaling pathway, involving the Janus Kinase (JAK) family and transcription factors STAT1/2. Activated STAT1 and STAT2 form a complex, leading to the induction of hundreds of IFN-stimulated genes (ISGs) that limit influenza viral replication [26].

Influenza viruses were shown to utilize several viral proteins to block the RIG-I-MAVS-mediated IFN-I production in many ways [28], one of which is inducing the degradation of molecules critical for IFN-I synthesis (Figure 2). More and more reports have shown that the influenza vRNP complex plays an important role in regulating innate immune response during infection. These inhibitory effects are at least partially due to virus-induced degradation of the mitochondrial adaptor MAVS by two viral proteins, NP and PB1. Bo Zhang et al. have revealed that the NP of the H1N1 virus (PR8 strain) was capable of associating with MAVS, as well as the cargo receptor-interacting protein called TOLLIP, which is involved in intracellular trafficking on mitochondria. This specific association induces the mitophagy of MAVS, leading to the degradation of this protein, thereby blocking MAVS-mediated production of IFNs [33]. The authors further identified Tyrosine313 of NP as essential for mediating mitophagy. Besides NP, another vRNP component, PB1, also displayed a regulatory effect on MAVS. PB1 of the H7N9 virus specifically interacted with and destabilized MAVS [34]. Further studies revealed that PB1 promoted the action of RNF5, which is an E3 ligase, to catalyze K27-linked polyubiquitination of MAVS at Lys362 and Lys461. PB1 was also shown to interact with a selective autophagic receptor, NBR1, which can recognize the ubiquitinated MAVS. Thereby, PB1 delivers MAVS to autophagosome for degradation, consequently blocking the RIG-I-MAVS-mediated antiviral innate immunity and facilitating H7N9 virus propagation [34]. Other studies demonstrated that viral PB1-F2 protein is capable of inhibiting the IFN-I response by targeting MAVS and decreasing mitochondrial membrane potential [35,36], although it is unclear whether PB1-F2 regulates MAVS protein level or not.

Besides MAVS, some other key adaptors in the IFN-I production pathway have also been proven to be downregulated during influenza virus infection. Ouyang Wei et al. showed that IAV (PR8 strain) infection increased the expression of N-Myc and STAT interactor (NMI), which interacted with IRF7, resulting in K48-linked ubiquitination and proteasomal degradation of IRF7. They further identified TRIM21 as the E3 ligase for IRF7. NMI utilized TRIM21 to promote the ubiquitination of IRF7 during IAV infection. However, the in-depth mechanism of how IAV manipulates NMI during this process remains unclear [37].

There are also cellular proteins with multiple functions that are not, in essence, classified as the conventional transductors of the IFN-I cascade. However, they are considered to at least partially participate in the IFN-I regulatory network during infection and positively affect IFN-I-mediated antiviral response by activating IFN transductors. Influenza viruses are likewise capable of acting on this type of protein, promoting their degradation to antagonize IFN-I signaling. Some of these regulatory proteins are cellular enzymes. For example, sphingosine 1-phosphate (S1P) lyase (SPL) is an intracellular metabolic enzyme that catalyzes the degradation of the bioactive lipid S1P. SPL has been reported to function in many cellular or disease processes, such as inflammation, cancer, and immunity. We have shown that overexpression of SPL suppressed IAV replication in cell culture [38]. Our research revealed that the antiviral function of SPL is closely related to the IFN-I signaling pathway. SPL interacts with IKKε, but not TBK1, to promote the IAV-induced IKKε activation, thereby increasing the synthesis of type I IFNs. Overexpression of SPL elevated the production of IFN-β and suppressed IAV replication. On the contrary, in SPL knockout cells, the IFN-β production in response to viral infection was significantly impaired [39]. Our recent research revealed that IAV counteracts the SPL-mediated antiviral effect by inducing SPL protein degradation. We determined that SPL was polyubiquitinated upon IAV infection. Further, viral NS1 was shown to induce SPL ubiquitination, followed by degradation of this protein. Consistent with the result, the NS1-deficient IAV failed to elicit SPL degradation [40,41]. These findings unravel an influenza viral strategy to subvert the SPL-mediated antiviral effect, which is to downregulate SPL by triggering its degradation.

Similar to ubiquitinases, certain de-ubiquitylases (DUBs) also regulate signaling pathways critical for host immunity and viral pathogenesis. A deubiquitylase called OTUB1 was proven to be a key regulator of RIG-I-dependent signaling, as it activates RIG-I via a coordinated mechanism of hydrolyzing K48 polyubiquitin chains attached to RIG-I and forming an E2-repressive complex [42]. Akhee Sabiha Jahan et al. have demonstrated that OTUB1 was induced by IFN-I but downregulated following IAV infection. The authors probed the crucial role of OTUB1 in maintaining the RIG-I mediated signaling. They utilized OTUB1-deficient cells to reveal that deletion of OTUB1 resulted in impaired IRF3 and NF-κB activation. Importantly, they further demonstrated that OTUB1 was targeted by IAV NS1 for protein degradation. NS1 proteins from a wide range of influenza A virus strains were shown to trigger the proteasomal degradation of OTUB1, thus antagonizing the RIG-I signaling cascade [42].

In some cases, influenza virus-triggered degradation of host IFN regulators is dependent on virus type or subtype. DEAD box protein 3 (DDX3) is a multifunctional protein involved in various processes linked to gene expression [43]. Since DDX3 was reported to interact with IKKε and TBK1, it contributes to the induction of antiviral type I IFNs [44,45,46]. Eun-Sook Park et al. have demonstrated that the PB1-F2 protein from the influenza A/Brevig Mission/1/1918 (H1N1) strain, but not from the PR8 strain, was able to interact with DDX3, leading to the proteasomal degradation of DDX3. This strain-specific binding and degradation of DDX3 further inhibits IFN-β induction during the 1918 influenza virus infection, at least partially being responsible for the high virulence of this strain [44,47].

Tumor necrosis factor receptor (TNFR)-associated factors (TRAFs) are intracellular signaling molecules that play important roles in regulating the host immune system. TRAF6 is a well-defined signaling molecule that interacts with MAVS and regulates IFN induction [48,49]. Zhaoshan Chen et al. have reported that the M1 protein of influenza D virus (IDV) inhibits the IFN-I response by interacting with TRAF6, mediating the K48-linked ubiquitination of TRAF6, followed by the degradation of this protein [50]. They further identified the E3 ligase KEAP1 as a ubiquitinase that promotes the ubiquitination of TRAF6 [50]. However, there is no evidence that the inhibitory effect of viral M1 is universal among different types of influenza viruses. The differential regulation of cellular proteins by different virus types/subtypes may account for the diversity of virulence or host range, which warrants further investigation.

### 3.2. Viral Degradation of Proteins in the IFN-I Receptor Signaling

Although influenza viruses strive to regulate crucial factors in the IFN-I production pathway, type I IFNs were still detectable clinically or experimentally [51,52]. Hence, apart from intervening in the IFN-I production pathway, it is reasonable that the viruses develop additional strategies to antagonize IFNAR-mediated antiviral signaling. Studies have shown that influenza viruses were capable of evading the IFNAR-mediated JAK-STAT signaling by eliminating crucial factors within this pathway (Figure 2). We have identified that IAV induces the reduction of IFNAR1 at the protein level [53]. Moreover, we determined the viral HA triggers the phosphorylation and both K48 and K63-linked polyubiquitination of IFNAR1, leading to the degradation of this receptor. Further, HA proteins from different IAV subtypes showed similar inhibitory effects on IFNAR1 levels [53]. We then investigated the cellular proteins that are functionally involved in this process. CK1α and PARP1 were proven to be critical for IAV HA-induced IFNAR1 degradation. Since the level of IFNAR1 on the cell surface is thought to be important for transmitting the signal of IFN-I, virus-induced degradation of IFNAR1 could extensively attenuate the cellular responses to IFNs. Indeed, both CK1α and PARP1 were shown to positively regulate viral replication by impairing IFN-I receptor signaling [54,55].

The JAK-STAT cascade downstream of IFNAR is also subverted by influenza viruses. Many reports have shown influenza viral regulation of JAK1, which is a key factor in transducing diverse signaling downstream to different receptors. Hui Yang et al. initially showed that IAV PB2 protein targets mammalian JAK1 at lysine 859 and 860 for ubiquitination and degradation [56]. Notably, the H5 subtype of highly pathogenic IAV with I283M/K526R mutations in PB2 enhanced the ability to degrade JAK1, which allows the virus to replicate with higher efficiency in mammalian cells but not in avian cells [56]. Shortly afterward, the same group established another discovery where they demonstrated the PA protein from the H5 subtype IAV is also capable of interacting with JAK1 [57]. Specifically, PA induced the K48-linked polyubiquitination of JAK1 at Lysin249. Importantly, the PA protein harboring 32T/550L degraded both mammalian and avian JAK1, while the PA harboring 32M/550I degraded only avian JAK1 [57]. Moreover, Yinping Du et al. from another group reported that IAV infection induces the expression of SOCS1, which, in turn, promotes ubiquitination and proteasome-dependent degradation of JAK1 [58].

In addition to PB2 and PA, the influenza virus uses viral NS1 to target the JAK-STAT signaling. Shi Liu et al. have found that IAV NS1 interacts with DNMT3B, which is one of the DNA methyltransferases [59]. NS1 induces the dissociation of DNMT3B from the promoter regulatory regions of the essential regulator of JAK-STAT signaling and then transports DNMT3B into the cytoplasm, resulting in the ubiquitination and degradation of this protein [59].

### 3.3. Viral Degradation of the Cellular Factors in the Type II and Type III IFN Pathways

IFN-γ is secreted by specific immune cells, such as T lymphocytes and natural killer (NK) cells, which bind to the cognate receptor (IFNGR) complex to elicit a signal in the infected cells [60]. The binding of IFN-γ with IFNGR activates JAK1 and JAK2 and causes phosphorylation of STAT1. The phosphorylated STAT1 forms a dimmer and translocates to the nucleus to associate with GAS elements (IFN-γ-activated site), resulting in the induction of IFN-γ-inducible genes [61,62]. IFN-γ has been shown to be important for both innate and adaptive immunity against different viral infections, including influenza virus [63,64,65,66,67]. IFN-γ is capable of inducing the expression of diverse factors that inhibit viral replication by impairing the accumulation of viral-specific mRNA, dsRNA, and protein. It can also inhibit cell growth and upregulate the expression of IRF-1, ICAM-1, and MHC molecules that influence virus propagation via stimulation of the host immunity [61]. MHC class II molecules play a critical role in antiviral immunity by presenting peptides derived from extracellular pathogens to CD4 T cells. The IFN-γ-induced MHC class II expression requires the JAK-STAT pathway [68,69].

Type III IFNs (IFN-λ1~λ4) are a newly identified type of molecules that are critical in controlling viral, bacterial, and fungal infections [70,71]. IFN-λ acts through the engagement of its cognate receptor complex, consisting of IFNLR1 and IL-10R2. STAT1 and STAT2 are activated by JAK1 and TYK2 and then transported to the nucleus to mediate the induction of IFN-λ-stimulated genes to regulate the host defense against infections [71].

Influenza viruses have evolved mechanisms to evade the IFN-γ and IFN-λ dependent anti-viral responses (Figure 2). Uetani Kohsaku et al. have reported that the IFN-γ-inducible expression of HLA-DRα, CIITA, ICAM1, IRF-1, and GBP1 were all inhibited in IAV-infected cells [72]. IAV infection also inhibited the IFN-γ-triggered nuclear translocation of STAT1. Mechanistically, IAV blocks the phosphorylation of STAT1 on Tyr701, which is required for the nucleus translocation of STAT1. Further, they demonstrated that the influenza virus markedly decreased the protein levels of JAK1 and IFNGR1, while the mRNA levels of each protein were not altered [72]. They presumed this may contribute to the extensive suppression of IFN-γ signaling cascade. However, the underlying mechanisms were not determined. We have reported that the influenza virus regulates IFN-γ signaling by inducing the degradation of IFN-γ receptor IFNGR1 via vial HA. Cellular protein casein kinase 1α (CK1α) was critical for IAV-induced degradation of this receptor [55]. Consequently, IAV infection extensively impairs cellular sensitivity to type II IFNs, which was experimentally demonstrated by the measurements of the IFN-γ triggered mRNA expression of TAP-1 and LMP-2, as well as protein levels of IRF-1 and phosphorylated STAT1 [55].

As for the type III IFN system, MuChun Tsai et al. have demonstrated that the receptor subunit IFNLR1 in human airway epithelia was rapidly degraded during influenza virus infection [73]. They have also identified the E3 ligase of IFNLR1, which is the Skp-Cullin-F ligase subunit FBXO45. They have shown that influenza viral infection induced the expression of FBXO45, which in turn mediated polyubiquitination of IFNLR1, followed by proteasome-dependent degradation of this protein, resulting in an impaired type III IFN signaling. Further, the Lysine319 and Lysine320 of IFNLR1 were shown to be important for the virus-induced ubiquitination of IFNLR1 [73].

As summarized in Section 3.2 and Section 3.3, several groups have reported the degradation of JAK1 by influenza virus infection [56,57,58,72]. Since JAK1 is important to transduce type I, type II, and type III IFN signaling, influenza viral dedication to vigorous degradation of JAK1 may effectively subvert all three types of interferon pathways at the same time, comprehensively countering the host innate and adaptive antiviral immunity.

## 4. Viral Degradation of Cellular Proteins That Do Not Directly Regulate IFN Responses

The IFN-I, IFN-II, and IFN-III signaling pathways are indeed the elemental systems by which the host restricts influenza virus infection. Nevertheless, there are also other cellular proteins that negatively regulate influenza virus infection by targeting different stages of the viral life cycle through multifarious mechanisms. These cellular proteins have unique special tactics to limit influenza virus propagation on their own. Additionally, some host proteins’ functions seem to interfere with influenza virus replication, although they are not particularly categorized as antiviral proteins. However, the influenza virus appears to have developed methods to instruct them to undergo the protein degradation pathways, mostly using viral proteins (Figure 3). The following contents are classified based on the regulatory functions of viral components to exemplify the cellular proteins subjected to the degradation.

### 4.1. Viral RNA Polymerase

Influenza viruses are able to induce ubiquitin-dependent degradation of essential enzymes in the mRNA synthesis machinery to broadly downregulate cellular gene expression [11,74]. Much evidence has implied that influenza viral RNA polymerase is associated with host RNA polymerase II (Pol II) and mediates the degradation of Pol II [74,75,76,77]. An increased ubiquitination of Pol II was detected during influenza virus infection or upon expression of the viral RNA polymerase. It was further determined that the expression of the PA subunit resulted in the decrease of Pol II. Studies with reassortant viruses indicated that PA and PB2 subunits individually contribute to Pol II degradation [75,78]. However, when infected with certain strains of IAV, such as A/PR/8/34 and A/Ann Arbor/6/60, the degradation of Pol II did not occur, suggesting the presence of strain specificity in the regulation [78]. C. M. Llompart et al. characterized two specific residues located in the PA and PB2 polymerase subunits of multiple influenza viruses that mediate this degradation [75]. These findings highlight a mechanism by which the influenza virus inhibits the host gene expression, which could have important implications for viral propagation [74].

Influenza viral RNA polymerase complex seems to have an additional function in altering host cellular proteins. Roberto Alfonso et al. have documented that influenza virus, but not other viruses such as VSV or adenovirus, specifically induces the degradation of CHD6, which is a chromatin remodeler protein both in vitro and in vivo [79]. CHD6 was previously shown to interact with influenza viral polymerase complex, repress viral replication, and re-localize to inactive chromatin during influenza virus infection [80]. The authors further demonstrated that the degradation of CHD6 is not dependent on the cellular proteasome pathway, and sole expression of any one of the three viral polymerase subunits is sufficient to induce the proteolysis of CHD6 [79].

The histone deacetylases (HDACs) are a group of enzymes that negatively regulate the cellular acetylation levels to control gene expressions. Several HDACs have been found to negatively affect IAV propagation, which include HDAC1 [81], HDAC2 [82], HDAC4 [83], HDAC6 [84], and HDAC11 [85]. IAV has also developed strategies to effectively antagonize their antiviral effects by downregulating the levels of these enzymes, mostly via viral RNA polymerase. For example, while HDAC6 plays crucial roles in many cellular processes, it is known to display anti-microbial function against various viruses and bacteria [86]. Specifically, HDAC6 binds to and deacetylates IAV RNA polymerase subunit PA, thereby inducing the proteasomal degradation of PA [87]. It has been documented that both the HDAC6-encoding mRNA and protein levels were downregulated during infection of A549 cells by multiple IAV strains [88]. HDAC6 polypeptide was shown to be cleaved by lysosome-associated caspases. Interestingly, influenza viral PA was determined to be implicated in the downregulation of HDAC6, which benefits the virus [88], suggesting the mutual constraints between HDAC6 and PA. Another HDAC, HDAC4, which was also reported to display anti-IAV properties, has been shown to be downregulated during IAV infection at both mRNA and protein levels, and this reduction was further proven to be modulated by viral PA-X [83]. In addition, IAV infection also inhibits the activity of some other HDACs, including HDAC1, HDAC2, and HDAC11, causing reductions of these HDACs at either mRNA level (HDAC11) or both mRNA and protein levels (HDAC1 and HDAC2) [81,82,85]. Further study indicated that IAV promotes the degradation of HDAC1 and HDAC2 polypeptides through the proteasome pathway [81,82]. However, the viral component(s) that is involved in the degradation of these HDACs were not determined in the research.

### 4.2. Viral NS1

Alternation in host cellular translation machinery is an intriguing area of research for influenza-host interaction [89]. During the protein synthesis stage of the life cycle, the influenza virus is able to selectively translate its own RNAs using host translational machinery while inhibiting cellular mRNA translation [90]. These strategies frequently involve the viral regulation of eukaryotic translation initiation factors (eIFs), which control the initiation mechanism of translation [89]. Song Wang et al. have demonstrated that IAV infection dramatically reduced the level of eIF4B, which belongs to the eIF family and is a crucial factor in regulating mRNA translation initiation [91]. They have demonstrated that eIF4B plays a vital role in the regulation of viral replication. Meanwhile, IAV infection diminishes eIF4B by inducing the lysosomal degradation of this protein both in cultured cells and in mice. Further, they identified viral NS1 as the main factor that mediates eIF4B instability. Moreover, they demonstrated that eIF4B positively regulates the expression of IFITM3, which is an important antiviral protein, to influence the replication of the influenza virus [91].

After the protein synthesis stage, the newly manufactured viral components, including vRNPs, must be packaged into progeny virion. A cellular protein called MOV10 was reported to interact with influenza viral NP and sequester the viral RNP in the cytoplasm, thereby displaying antiviral activity [92,93,94]. However, influenza viral NS1 was shown to be associated with MOV10 [93,95]. Jian Li et al. further investigated this interaction and demonstrated that NS1 antagonizes the antiviral function of MVO10 by both reducing the association between MOV10 and NP and inducing the degradation of MOV10 through the lysosome-dependent pathway [93].

As a transcription factor, p53 is initially known for its role as a tumor suppressor, modifying gene expression such as BAX to regulate the fate of a cell. It is also involved in many biological processes [96,97,98]. It has been reported that p53 displays antiviral activity against influenza viruses by affecting both innate and adaptive immunity [99]. For instance, p53 can upregulate IRF9; thus, it promotes IFN-mediated antiviral immunity. Other genes encoding proteins involved in innate immunity, such as TLR3 [100], IRF5 [101], PKR [102,103], and MCP-1 [104], are also p53’s direct transcriptional targets. p53 is also involved in pulmonary monocyte infiltration, as well as T cell and dendritic cell responses to influenza virus infections [99]. Oliver Terrier et al. have shown that during infection of A549 cells with diverse influenza A virus subtypes (H3N2, H5N1, H5N2, or H7N1), there were global down-regulation of genes both upstream and downstream of p53, such as AKT1, PTEN, p21, PERP, FAS, DR4/5, BAX, Bcl-XL, PAI-I, and so on, attenuating the entire p53 cascade [105]. However, the mRNA and protein levels of p53 itself remained stable or even upregulated during infection [105]. In the p53 signaling, there is an important regulator, MDM2, which is essentially an E3 ubiquitin ligase and is considered a key negative regulator of p53. MDM2 directly binds to p53 and promotes the ubiquitination and proteasomal degradation of p53 [106,107]. MDM2 also mediates NEDDylation of p53 [108]. Pizzorno Andres et al. have investigated the role of MDM2 during IAV infection. The results indicate that the MDM2 protein was markedly degraded at an early stage of viral infection [109]. IAV also altered the nuclear-cytoplasmic localization of MDM2. Furthermore, Viral NS1 was proven to contribute to IAV-induced MDM2 degradation. By utilizing siRNA-based knockdown or exogenous expression methods, MDM2 was proven to display antiviral activity, which is independent of its E3 ligase activity [109]. Influenza viral degradation of MDM2 should provide a cellular environment favorable for virus propagation. However, the detailed molecular mechanism remains to be further investigated.

Yan Zhao et al. have reported a novel mechanism employed by highly pathogenic avian IAV to further escape host innate immunity, which is through altering the transcriptional readthrough (TRT) [110]. The authors showed that NS1 binds to SSU72 (RNA polymerase II subunit A C-terminal domain phosphatase), leading to a reduction in its expression that eventually induces TRT and represses STAT1/2 mRNA expression, which enabled the virus to evade host antiviral immune responses [110]. IAV infection was also shown to induce the degradation of a sub-nuclear structure known as the Nuclear Domain-10 (ND-10) [111], which has been known to play a role in antiviral defense during IAV infection [112,113]. Infection with IAV or expression of NS1 in A549 cells degraded the main component of the ND-10 anti-viral complex, PML. A follow-up study showed that NS1 induced SUMOylation of the PML nuclear body, leading to the disintegration of the host ND-10 complex [111].

### 4.3. Viral M2

Influenza viral M2 is an ion channel protein that has been widely studied for its role in regulating host pathways/proteins. James David Londino et al. have reported that M2 induces the degradation of a cellular protein, cystic fibrosis transmembrane conductance regulator (CFTR) [114]. CFTR influences cytokine responses and plays an important role in viral infection-associated inflammation and mortality. The authors attempted to understand the expression and functional alteration of CFTR during influenza virus infection. Indeed, IAV infection significantly reduced the total level and level of CFTR expressed on the plasma membrane via lysosome-dependent degradation of CFTR [114]. Furthermore, IAV M2 protein was proven to be crucial for CFTR degradation during infection, as either siRNA-based M2 knockdown or inhibition of M2 activity abolished M2-induced degradation of CFTR. Due to the multiple functions of CFTR in cytokine expression, antioxidant response, epithelial integrity, etc., exactly how the degradation of CFTR benefits the influenza virus remains to be further explored [114]. Furthermore, another cellular binding partner of influenza viral M2 has been identified by a yeast two-hybrid screen, which is cyclin D3, a key regulator of the cell cycle [115]. Ying Fan et al. determined a novel antiviral activity of cyclin D3, as silencing of this gene resulted in a significant increase in influenza viral progeny titers in the supernatant of infected cells. Further, they demonstrated that the antiviral effect of cyclin D3 is due to its interference with viral M1-M2 interaction, which is essential for the proper assembly of viral particles. Importantly, influenza virus infection altered the distribution of cyclin D3 from the nucleus to the cytoplasm, followed by proteasomal degradation of this protein. However, the detailed mechanism of how cyclin D3 is targeted for degradation and which E3 ligase catalyzes the ubiquitination of cyclin D3 remains unknown [115].

### 4.4. Viral NS2 and NA

Although there were relatively fewer reports, the other influenza viral proteins, such as NA and NS2, have also been shown to trigger the degradation of host factors. G protein pathway suppressor 2 (GPS2) is a cellular protein involved in the RAS/MAPK pathway and displays inhibitory functions in cellular or viral transcription [116,117]. By performing a yeast two-hybrid screening assay, Wenxiao Gong et al. have identified GPS2 as a binding partner for influenza viral NS2 (NEP) [118]. Knockdown or knockout of GPS2 enhanced IAV titers, whereas overexpression of GPS2 suppressed viral replication, suggesting GPS2 as a host antiviral factor. In principle, GPS2 inhibited viral RNA synthesis by reducing the assembly of viral polymerase [118]. The authors further demonstrated that IAV NS2 interacts with GPS2 and mediates its nuclear export, inducing the degradation of this protein, therefore suppressing GPS2-mediated inhibition of viral polymerase function [118].

Influenza viral NA also interacts with the host and displays additional activity besides its original neuraminidase function. Xiangwu Ju et al. have demonstrated that NA of H5N1 IAV triggers deglycosylation and degradation of lysosome-associated membrane proteins (LAMPs), which induces lysosomal rupture, leading to the death of A549 cells and human tracheal epithelial cells [119]. NA protein of IAV was also shown to regulate the cleavage of angiotensin-converting enzyme 2 (ACE2), which was well-established as a functional receptor for both SARS coronavirus (SARS-CoV) and SARS-CoV-2 [120,121], resulting in the protein degradation of ACE2 through the proteasome pathway [122]. However, the ACE2 downregulation did not seem to affect IAV replication in vitro, and the relevance of ACE2 cleavage by NA in influenza viral pathogenesis needs further investigation [122].

### 4.5. Yet-Unidentified Viral Component

In addition to the lists summarized above, there are still other cellular proteins that are degraded during influenza virus infection caused by a yet-unknown viral component. These proteins participate in many cellular pathways and display a negative impact on influenza viral replication. Recently, IAV proteins, including NP, NS1, NA, M1, and PB1-F2, have been implied to be involved in the cell apoptotic process [123,124,125,126,127,128]. In fact, inducing the early apoptosis of host alveolar macrophages (AMs) could be one of the most effective strategies employed by IAV to facilitate its survival [129]. PB1-F2 was shown to induce apoptosis in macrophages by targeting mitochondria, which could be inhibited by a cellular protein NLRX1 (nucleotide-binding oligomerization domain-like receptor X1), a recently recognized member of the NOD-like receptors [130]. Mengyuan Cen et al. have demonstrated that during infection, IAV hijacks an F-box protein, FBXO6, to promote K48-linked ubiquitination and proteasomal degradation of NLRX1, which can then facilitate IAV-induced apoptosis of AMs and impair the AM-mediated antiviral immunity [131]. Nevertheless, the viral component that is directly involved in this process is currently unclear.

To understand the mechanism of how the highly pathogenic H5N1 IAV infection leads to acute respiratory distress syndrome (ARDS) [132], Tao Ruan et al. studied the modification of intercellular junction proteins observed upon H5N1 infection [133]. Their results showed that H5N1 viruses downregulate the expression of junction proteins such as occludin, E-cadherin, claudin-1, and ZO-1. Further studies revealed that H5N1 infection activates the E3 ligase Itch to promote the ubiquitination of occludin, followed by its degradation. Their research highlights a plausible mechanism by which the highly pathogenic IAV impairs the alveolar epithelial barrier to cause severe diseases [133].

Influenza viruses can also differentially regulate host proteins according to different infection stages. For instance, Da-Yuan Chen et al. reported that a cellular actin filament-binding protein, cortactin, promotes IAV infection at an early time point but undergoes degradation via a lysosome-dependent pathway during the late stages of IAV infection [134,135]. This may be due to an inhibitory role of cortactin during the late stages of IAV infection, which decreases the amount of viral progeny released from infected cells. Thus, IAV facilitates its degradation in order to undermine such function [134]. IAV infection has been identified as one of the major causes of ROS generation, which may lead to the activation of acute or sustained oxidative stress [136]. The impact of oxidative stress on IAV infection is likely to benefit viruses at a certain stage of their life cycle [137]. Kwang Il Jung et al. have demonstrated that the early stage of IAV infection induces autophagic degradation of antioxidant enzyme SOD1 through the lysosome-dependent method, thereby contributing to increased ROS generation and viral infectivity in alveolar epithelial cells [138].

## 5. Conclusions and Perspectives

Upon influenza virus infection, dynamic interactions between virus and host take place where many cellular proteins can be manipulated at the level of either expression or biological activities. Some of the alterations are due to the host’s antiviral response to the infection, while some others can be directly instigated by the viruses to fulfill their robust replication and propagation.

As reviewed above, a considerable number of cellular proteins that regulate IFN responses have been shown to be degraded by the influenza virus (Table 1). This is reasonable because the viruses are dedicated to countering the host’s antiviral response to create an environment conducive to their own replication, and the IFN-I signaling is the host’s most powerful strategy to limit viral replication. Indeed, the recombinant IFN-I has been clinically utilized or tested for treating many virus-related diseases. For example, IFN-α has been used for the treatment of infections by hepatitis B virus (HBV), HEV, and severe acute respiratory syndrome [139,140]. In the case of HCV infection, recombinant IFN-α had long been used as a therapy until the viral protease-specific drug was developed. However, the antiviral therapeutic activity of IFNs is largely attenuated by viral escape mechanisms, which greatly limits the effectiveness of IFNs for the treatment of various viral infections. Further, IFN-I could negatively regulate the host immunity and even be pathogenic to the host in some conditions [141]. Therefore, identification of the specific IFN regulatory proteins degraded by the virus may not only increase our understanding of virus-host defense interplay but also help to improve designing new therapeutics against virus infections.

The ubiquitin system is one of the most potent post-translation modifications (PTMs) that control the activation or inactivation of various proteins. The roles of ubiquitin in innate immunity and its implications in the IAV have been well established [142,143]. There are E1 (ubiquitin-activating enzymes), E2 (ubiquitin-conjugating enzymes), and numerous E3 (ubiquitin-ligase enzymes) that sequentially catalyze the ubiquitination process of a target protein, thereby either regulating its function or inducing the degradation. As summarized in Section 3 and Section 4, several E3 ligases have been revealed to mediate the degradation of cellular proteins upon IAV infection, but many others remain unknown. Identification of E3 ubiquitin ligases that are activated upon infection could be the key step for defining the molecular mechanisms for influenza virus-induced destruction of cellular antiviral proteins. Furthermore, the ubiquitinase or the related downstream cellular factors may potentially represent exciting target molecules for instituting the host protein-directed therapeutics curing viral diseases such as influenza.

## Figures and Tables

**Figure 1 ijms-25-04677-f001:**
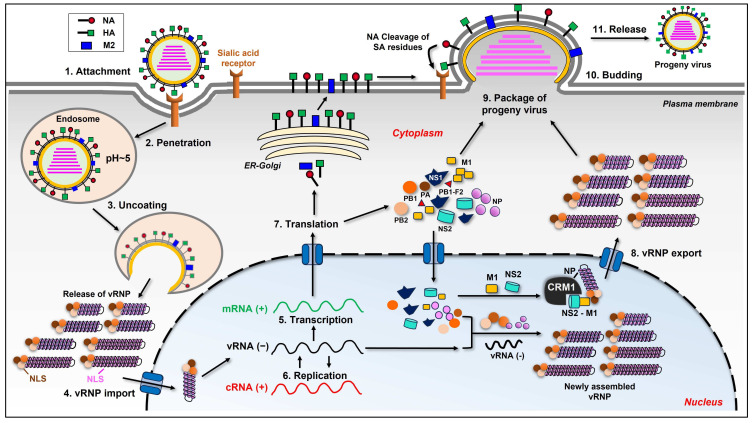
The life cycle of influenza virus. The life cycle is depicted as 11 stages from virus attachment to release in the figure. The virion first attaches to the host cell with the binding of viral HA to the sialic acid receptor on the cell surface. The virion then penetrates the cell via endocytosis. A low pH value inside the endosome triggers the fusion of the endosomal membrane with the viral envelope, releasing the viral ribonucleoprotein (vRNP) into the cytoplasm, which is termed “uncoating”. The eight vRNPs then enter the nucleus with the help of nuclear localization signals (NLS) existing in each component of the vRNP complex. After entering the nucleus, the negative sense viral RNA [vRNA(−)] is converted into a positive sense RNA [cRNA(+)] to serve as a template for the replication of progeny vRNA(−). The virus also utilizes the host’s transcriptional machinery to generate its own mRNA(+). The mRNAs translocate to the cytoplasm and are translated into at least 11 viral proteins. Three viral proteins found within the viral envelope, which are HA, NA, and M2, are transported to the cellular plasma membrane through ER Golgi. Viral proteins, such as PA, PB1, PB2, and NP, are imported into the nucleus to form progeny vRNP complex together with the newly synthesized vRNA(−). The newly assembled vRNPs are then exported from the nucleus to the cytoplasm with the help of M1, NS2, and NP. After vRNP export, the newly generated viral components (proteins and vRNP) are packaged into progeny virions and budding from the host cell. During this step, the virus utilizes the host cell’s plasma membrane to form its own envelope, which contains viral HA, NA, and M2 on the surface. Lastly, viral NA must cleave the sialic acid residues from glycoproteins and glycolipids, helping the viral particles release from the host cell.

**Figure 2 ijms-25-04677-f002:**
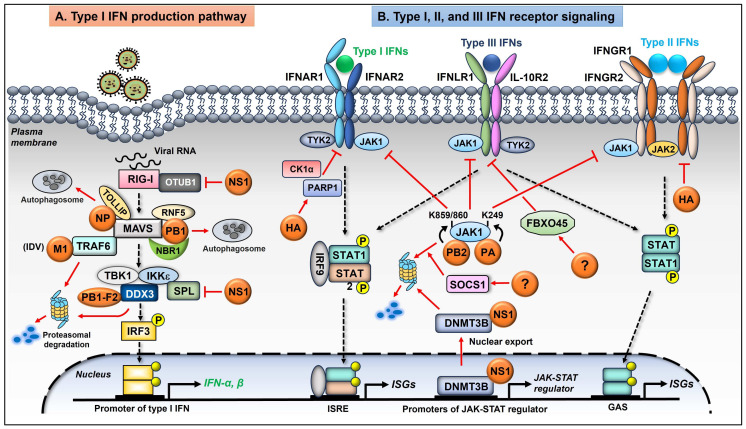
Schematic representation of the influenza viral degradation of cellular proteins involved in the interferon responses. (**A**) During influenza virus infection, the RIG-I-mediated signaling pathway is activated to promote the production of type I IFNs. Viral NS1 targets OTUB1, which is a key regulator of RIG-I for proteasomal degradation, therefore antagonizing the RIG-I activation. Viral NP and PB1 interact with MAVS, leading to the degradation of MAVS through autophagosomes with distinctive mechanisms, thereby blocking the MAVS-mediated activation of the downstream factors. The M1 protein of IDV interacts with TRAF6, which positively regulates MAVS and mediates the K48-linked ubiquitination of TRAF6, triggering the degradation of this protein. IAV NS1 induces the ubiquitination and degradation of SPL, resulting in an impaired activation of IKKε. PB1-F2 protein from influenza A/Brevig Mission/1/1918 (H1N1) strain binds to DDX3, which contributes to type I IFN induction, leading to the proteasomal degradation of this protein. (**B**) Influenza viral HA triggers the phosphorylation and ubiquitination of IFNAR1, leading to the degradation of this receptor. Cellular proteins CK1α and PARP1 are critical for HA-induced IFNAR1 degradation. IAV PB2 protein targets JAK1 for ubiquitination at lysine 859 and 860 (K859/860), while PA from H5 subtypes of IAV interacts with JAK1 and induces ubiquitination of JAK1 at K249. Both viral proteins are capable of triggering the ubiquitination-dependent degradation of JAK1, leading to impaired signaling transduction for all three types of interferons (type I, II, and III). IAV also upregulates SOCS1, which promotes proteasomal degradation of JAK1. Influenza viral NS1 interacts with DNA methyltransferase, DNMT3B, and induces the dissociation of DNMT3B from the promoter regulatory regions of the JAK-STAT regulator, transporting DNMT3B into the cytoplasm, followed by the ubiquitination and proteasomal degradation of this protein. Influenza virus infection upregulates the expression of an E3 ligase of IFNLR1 called FBXO45 by a yet-unidentified viral mechanism, which in turn mediates ubiquitination of IFNLR1, leading to the proteasome-dependent degradation of this receptor. Viral HA also induces the phosphorylation and ubiquitination of IFNGR1, resulting in the degradation of IFNGR1, thus suppressing the type II IFN-mediated antiviral defense.

**Figure 3 ijms-25-04677-f003:**
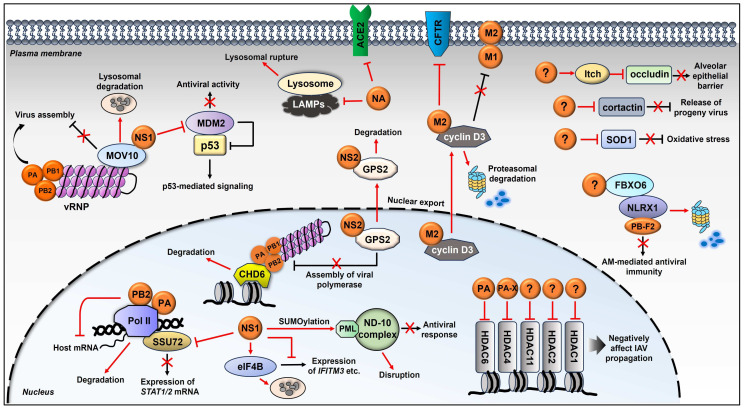
Schematic representation of the influenza viral degradation of cellular proteins that do not directly regulate IFN signaling. Schemes follow the same formatting. In the cytoplasm of the host cell, influenza viral NS1 interacts with MOV10 and induces the degradation of MOV10 through the lysosome-dependent pathway, therefore reducing the association between MOV10 and viral NP, facilitating the assembly of progeny virus. NS1 is also shown to trigger the degradation of MDM2, which is a negative regulator of p53 signaling, thereby suppressing the MDM2-mediated antiviral activity. NA of H5N1 IAV triggers de-glycosylation and degradation of LAMPs, inducing lysosomal rupture. NA also regulates the cleavage of ACE2 receptors, resulting in the degradation of ACE2 through the proteasome-dependent pathway. Viral M2 induces the degradation of the CFTR receptor, which plays an important role in virus-associated inflammation and mortality. Cellular protein Cyclin D3 interferes with viral M1-M2 interaction, which is essential for viral assembly. M2 interacts with cyclin D3, altering the distribution of cyclin D3 from the nucleus to the cytoplasm, followed by proteasomal degradation of this protein. In the nucleus, PA and PB2 subunits of influenza viral RNA polymerase complex associate with host RNA polymerase II (Pol II) and mediate the degradation of Pol II, resulting in the inhibition of host gene expression. The vRNP is shown to interact with CHD6, which represses viral replication. Each of the three subunits of the vRNP complex is capable of binding to CHD6, inducing the degradation of this protein. Viral NS1 interacts with SSU72 and elF4B, respectively, in the nucleus, inducing the degradation of both proteins, thus suppressing their antiviral function. NS1 also mediates SUMOylation of the main component of ND-10 complex, PML, leading to the disruption of ND-10, thus abolishing its antiviral activity. Cellular GPS2 protein inhibits IAV RNA synthesis by reducing the assembly of viral polymerase. Viral NS2 interacts with GPS2 and mediates its nuclear export, inducing the degradation of GPS2, thus abolishing its antiviral function. Host histone deacetylases (HDACs) display a negative effect on IAV propagation under certain conditions. Viral PA and PA-X induces the protein degradation of HDAC6 and HDAC4, respectively. The protein levels of HDAC1 and HDAC2, as well as the mRNA levels of HDAC1, HDAC2, and HDAC11, also decrease upon influenza virus infection by yet-unidentified mechanisms. The cellular NOD-like receptor NLRX1 binds to viral PB1-F2, thus preventing IAV-induced apoptosis in alveolar macrophages (AMs). IAV infection hijacks E3 ligase FBXO6 to promote the proteasomal degradation of NLRX1, facilitating the apoptosis of AMs and impairing AM-mediated antiviral immunity. H5N1 IAV activates E3 ligase Itch by an unidentified mechanism, thus promoting the ubiquitination of junction protein occludin, leading to its degradation. By inducing the degradation of occludin as well as other junction proteins, the highly pathogenic IAV, such as H5N1, impairs the alveolar epithelial barrier to cause severe disease. IAV facilitates the degradation of cortactin during the late stages of infection to undermine the inhibitory effect of cortactin on the release of progeny virus. At the early stage of infection, IAV induces lysosomal degradation of SOD1, thereby contributing to increased ROS generation and viral infectivity in alveolar epithelial cells.

**Table 1 ijms-25-04677-t001:** Influenza viral degradation of host cellular proteins.

Cellular Proteins	Viral Component	Function of the Cellular Proteins	Degradation Pathway	Relevant E3 Ubiquitin Ligase	Reference
OTUB1	NS1	Activates RIG-I via hydrolyzing K48-linked polyubiquitin chains of RIG-I	Proteasomal pathway	ND	[42]
MAVS	NP, PB1	A key adaptor for transducing RIG-I mediated signaling	Mitophagy pathway	RNF5	[33,34]
IRF7	ND	A transcription factor that is activated downstream TBK1/IKKε and promotes the induction of type I IFNs	Proteasomal pathway	TRIM21	[37]
SPL	NS1	Interacts with IKKε to promote the IAV induced type I IFN response	Proteasomal pathway	ND	[38,39,40,41]
DDX3	PB1-F2	Interacts with IKKε and TBK1 and contributes to the induction of type I IFNs	Proteasomal pathway	ND	[43,44,45,46,47]
TRAF6	M1 (IDV)	Interacts with MAVS and is involved in the activation of IRF3/IRF7	Proteasomal pathway	KEAP1	[48,49,50]
IFNAR1	HA	Binds to IFN-α/β and mediates the type I IFN antiviral signaling	Proteasomal and lysosomal pathway	SCF (HOS)	[53,54,55]
JAK1	PB2, PA (H5)	Transduces signal form diverse receptors and activates STAT1/2	Proteasomal pathway	ND	[56,57,58]
DNMT3B	NS1	Catalyzes DNA methylation affecting the expression of JAK-STAT regulators	Proteasomal pathway	ND	[59]
IFNGR1	HA	Binds to IFN-γ and mediates the type II IFN signaling pathway	Lysosomal pathway	ND	[55]
IFNLR1	ND	Binds to IFN-λ and mediates the type III IFN signaling pathway	Proteasomal pathway	FBXO45	[73]
RNA polymerase II	PA, PB2	An enzyme responsible for the transcription of mRNAs	ND	ND	[74,75,76,77,78]
CHD6	PA, PB1, PB2	Affects the binding of transcription factors thus modulating the initiation and elongation steps of transcription	ND	ND	[79,80]
HDAC1	ND	The most abundant member of the class I HDACs in pulmonary endothelial cells, regulating diverse cellular procedures	Proteasomal pathway	ND	[81]
HDAC2	ND	A class I HDAC that mostly targets histone H3 and H4 for deacetylation and plays important roles in multiple cellular events	Proteasomal pathway	ND	[82]
HDAC4	PA-X	A member of class IIa HDACs that is localized to both the nucleus and cytoplasm and has no deacetylase activity	Cleaved by lysosome-associated caspase	ND	[83]
HDAC6	PA	A class IIb HDAC that is non-nuclear and deacetylates cytoplasmic substrates such as tubulin.	Cleaved by lysosome-associated caspase	ND	[84] [86,87,88]
elF4B	NS1	Regulates mRNA translation initiation	Lysosomal pathway	ND	[89,90,91]
MOV10	NS1	Interacts with influenza viral NP and sequesters the vRNP in the cytoplasm	Lysosomal pathway	ND	[92,93,94,95]
MDM2	NS1	Negatively regulates p53 pathway by promoting the ubiquitination and degradation of p53	Proteasomal pathway	ND	[106,107,108,109]
SSU72	NS1	Involved in transcription termination process	ND	ND	[110]
ND-10	NS1	Involved in the replication of numerous viruses and in host cell responses to antiviral cytokines	SUMOylation mediated disruption	ND	[111]
CFTR	M2	Influences cytokine responses and plays an important role in viral infection-associated inflammation and mortality	Lysosomal pathway	ND	[114]
cyclin D3	M2	A key regulator of cell cycle	Proteasomal pathway	ND	[115]
LAMPs	NA	Present in the lysosome membrane and function to maintain the structural integrity of lysosomal compartment to prevent hydrolytic enzyme release	Deglycosylation mediated diminishment	ND	[119]
ACE2	NA	Serves as a functional receptor for both SARS-CoV and SARS-CoV-2	Proteasomal pathway	ND	[120,121,122]
GPS2	NS2	Participates in the RAS/MAPK pathway and displays inhibitory function in cellular or viral transcription	Proteasomal pathway	ND	[116,117,118]
NLRX1	ND	A pattern-recognition receptor belonging to the NLR family that localizes to the mitochondria	Proteasomal pathway	FBXO6	[130,131]
occludin	ND	One of the major components of the tight junction in epithelial and endothelial cells which is crucial for the alveolar epithelial barrier	Proteasomal pathway	Itch	[133]
cortactin	ND	A central regulator of branched filamentous actin network and is associated with the infection of various bacterial and viral pathogens	Lysosomal pathway	ND	[134,135]
SOD1	ND	An antioxidant enzyme that regulates cellular oxidative stress	Lysosomal pathway	ND	[136,137,138]

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
