# Peer review of "Triggering Degradation of Host Cellular Proteins for Robust Propagation of Influenza Viruses"

_ijms, 2024, doi:10.3390/ijms25094677_

Round 1

Reviewer 1 Report

Comments and Suggestions for Authors

The authors reviewed works related to degradation of host cellular proteins for robust propagation of influenza viruses. The MS is well prepared. Only a couple of comments listed below:

1.      Line 33-35, please modify the sentence.

2.      Line 86-89. More and more studies have been documented avian IAV could bind α(2,6) Sialic acids, as well as binding to Neu5Gc glycans as a different types of sialic acids as receptors for IAVs. Please briefly discuss this.

Author Response

Reviewer 1.

A. Comment: The authors reviewed works related to degradation of host cellular proteins for robust propagation of influenza viruses. The MS is well prepared.

Response: We greatly appreciate the positive evaluation on our manuscript.

B. Comment: Line 33-35, please modify the sentence.

Response: Thank you for noting the error in the sentence. We modified the sentence by deleting the repeated phrase.

C. Comment: Line 86-89. More and more studies have been documented avian IAV could bind α(2,6) Sialic acids, as well as binding to Neu5Gc glycans as a different types of sialic acids as receptors for IAVs. Please briefly discuss this.

Response: Thank you for the valuable information about the receptor usage. We have now included those studies on the use of α(2,6) sialic acids by some avian IAVs and Neu5Gc glycans by HAs in the text which are yellow highlighted in the revised manuscript.

Reviewer 2 Report

Comments and Suggestions for Authors

Thank you for the opportunity to review "Triggering degradation of host cellular proteins for robust propagation of influenza viruses' paperwork by Xia et al.

1. The article provides deep insight and systematic overview of influenza virus replication in the cell and the affected cellular stuctures and mechanisms by which the virus ensures its propagation and spread. The style is comprehensive and language is fine gramatically, as well. Citations are aduequate. Figures and tables are detailed and informative.Several host processes and participating factors, which ar targeted by the virus are discussed in a quite a comprehensive manner. The focus falls on ubiquitination as highlighted by the authors themselves. Thus a question arises why some key publications in the field are missing such as a review with 344 references by Lamotte and Tafforeau, 2021.  10.3390/v13112309)  , Rudnicka, A.; Yamauchi, Y., 2016 https://doi.org/10.3390/v8100293 and Park et al.,, 2022  https://doi.org/10.3390/ijms23094593. 

Another question regarding ubiquitination is why the authors selected to highlight exactly this process  as a potential promising target for antiviral approach. They stated their research in the field but are there any data on experimental therapeutics tested?

It appeared suprising to me not to see citations of recognised teams working on influenza infection and signalling pathways research. Such papers include those by Zhirnov O.P., Klenk H.-D., 2007; Hrincius E.R et al., 2011; Ehrhardt C. at al., 2007; Ludwig S. et al., 2002; García-Sastre A., 2011 and others. As an overall impression despite the detailed information and the volume, citations might be enriched.

2. There too may definite articles "the"used in particular expresions which might be reduced in some places such as "The vRNP complexes", "often hijack the cellular mechanisms.", " Human influenza viruses only recognize the α(2,6), whereas avian and equine influenza viruses recognize the α(2,3) linkage" and others. It would lighten the reading.

3. There are some concept repetiotions such as "In this review, we discuss influenza virus-induced degradation of cellular proteins, highlighting the modes of how the viruses co-opt the host by degrading these proteins to support viral propagation. We provide an overview of current knowledge about themechanisms employed by influenza virus to overcome the host antiviral response via the degradation of cellular proteins for robust replication and spread." The two sentences have the same meaning.

Despite those remarks I would recommend the review for publishing.

Comments on the Quality of English Language

There too may definite articles "the"used in particular expresions which might be reduced in some places such as "The vRNP complexes", "often hijack the cellular mechanisms.", " Human influenza viruses only recognize the α(2,6), whereas avian and equine influenza viruses recognize the α(2,3) linkage" and others. It would lighten the reading.

Author Response

Reviewer 2.

A. Comment: The article provides deep insight and systematic overview of influenza virus replication in the cell and the affected cellular structures and mechanisms by which the virus ensures its propagation and spread. The style is comprehensive, and language is fine grammatically, as well. Citations are adequate. Figures and tables are detailed and informative. Several host processes and participating factors, which are targeted by the virus are discussed in a quite a comprehensive manner.

Response: We greatly appreciate the positive evaluation on our review.

B. Comment: The focus falls on ubiquitination as highlighted by the authors themselves. Thus a question arises why some key publications in the field are missing such as a review with 344 references by Lamotte and Tafforeau, 2021.  10.3390/v13112309)  , Rudnicka, A.; Yamauchi, Y., 2016 https://doi.org/10.3390/v8100293 and Park et al.,, 2022  https://doi.org/10.3390/ijms23094593. 

Response: We feel sorry that the papers were missed in the prior manuscript. Those key papers are now added to the revised manuscript.

C. Comment: Another question regarding ubiquitination is why the authors selected to highlight exactly this process as a potential promising target for antiviral approach. They stated their research in the field but are there any data on experimental therapeutics tested?

Response: Since the ubiquitination and downstream degradation promote virus replication and viral propagation, the molecules involved in the process are considered good targets for the antiviral approach. Although we are unaware of any specific experimental therapeutics tested, we believe that it has great promise for future development. 

D. Comment: It appeared surprising to me not to see citations of recognized teams working on influenza infection and signalling pathways research. Such papers include those by Zhirnov O.P., Klenk H.-D., 2007; Hrincius E.R et al., 2011; Ehrhardt C. at al., 2007; Ludwig S. et al., 2002; García-Sastre A., 2011 and others. As an overall impression despite the detailed information and the volume, citations might be enriched.

Response: We agree that those citations could increase the quality of the review from the standpoint of influenza-host interaction studies. Thus, they are added to the revised manuscript.

E. Comment: There too may definite articles "the"used in particular expressions which might be reduced in some places such as "The vRNP complexes", "often hijack the cellular mechanisms.", " Human influenza viruses only recognize the α(2,6), whereas avian and equine influenza viruses recognize theα(2,3) linkage" and others. It would lighten the reading.

Response: Thank you for noting the point. We fixed it to increase the readability.

F. Comment: There are some concept repetitions such as "In this review, we discuss influenza virus-induced degradation of cellular proteins, highlighting the modes of how the viruses co-opt the host by degrading these proteins to support viral propagation. We provide an overview of current knowledge about the mechanisms employed by influenza virus to overcome the host antiviral response via the degradation of cellular proteins for robust replication and spread." The two sentences have the same meaning.

Response: We agree with the reviewer. Therefore, the first sentence was slightly modified, and the second sentence was deleted to eliminate the repetition and increase the readability.

G. Comment: Despite those remarks I would recommend the review for publishing.

Response: Thank you very much for the supportive comment.
